# Growth patterns in childhood and adolescence and adult body composition: a pooled analysis of birth cohort studies from five low and middle-income countries (COHORTS collaboration)

Natalia E Poveda [1], Linda S Adair [2], Reynaldo Martorell [1], Shivani A Patel [1], Manuel Ramirez-Zea [3], Santosh K Bhargava,[4] Sonny A Bechayda [5], Delia B Carba,[5] Maria F Kroker-Lobos,[3] Bernardo Lessa Horta [6], Natália Peixoto Lima,[6] Mónica Mazariegos [3], Ana Maria Baptista Menezes [6], Shane A Norris [7], Lukhanyo H Nyati,[7] Linda M Richter [8], Harshpal Sachdev [9], Fernando C Wehrmeister [6], Aryeh D Stein [1]

For numbered affiliations see end of article.

**Correspondence to**
Dr Aryeh D Stein;
aryeh.stein@emory.edu

## ABSTRACT

**Objective** We examined associations among serial measures of linear growth and relative weight with adult body composition.

**Design** Secondary data analysis of prospective birth cohort studies.

**Settings** Six birth cohorts from Brazil, Guatemala, India, the Philippines and South Africa.

**Participants** 4173 individuals followed from birth to ages 22–46 years with complete and valid weight and height at birth, infancy, childhood and adolescence, and body composition in adult life.

**Exposures** Birth weight and conditional size (standardised residuals of height representing linear growth and of relative weight representing weight increments independent of linear size) in infancy, childhood and adolescence.

**Primary outcome measures** Body mass index, fat mass index (FMI), fat-free mass index (FFMI), fat mass/fat-free mass ratio (FM/FFM), and waist circumference in young and mid-adulthood.

**Results** In pooled analyses, a higher birth weight and relative weight gains in infancy, childhood and adolescence were positively associated with all adult outcomes. Relative weight gains in childhood and adolescence were the strongest predictors of adult body composition (β (95% CI) among men: FMI (childhood: 0.41 (0.26 to 0.55); adolescence: 0.39 (0.27 to 0.50)), FFMI (childhood: 0.50 (0.34 to 0.66); adolescence: 0.43 (0.32 to 0.55)), FM/FFM (childhood: 0.31 (0.16 to 0.47); adolescence: 0.31 (0.19 to 0.43))). Among women, similar patterns were observed, but, effect sizes in adolescence were slightly stronger than in childhood. Conditional height in infancy was positively associated with FMI (men: 0.08 (0.03 to 0.14); women: 0.11 (0.07 to 0.16)). Conditional height in childhood

## STRENGTHS AND LIMITATIONS OF THIS STUDY

⇒ We analysed six well-characterised population-based prospective birth cohorts from five low and middle-income countries.

⇒ We considered a two-compartment model of body composition (fat mass index and fat-free mass index), that were measured with gold standards and/or validated techniques at each study site.

⇒ The use of conditional growth measures allowed us to break the high correlations among serial measures of weight and height to study the independent roles of linear growth and relative weight at specific age intervals.

⇒ Attrition might be one source of selection bias given the long-lasting follow-up of these birth cohorts.

⇒ Residual confounding by unmeasured characteristics such as diet, reproductive history and other lifestyle factors might bias the estimates.

was positively but weakly associated with women's adiposity. Site-specific and sex-stratified analyses showed consistency in the direction of estimates, although there were differences in their magnitude.

**Conclusions** Prenatal and postnatal relative weight gains were positive predictors of larger body size and increased adiposity in adulthood. A faster linear growth in infancy was a significant but weak predictor of higher adult adiposity.

## INTRODUCTION

Low and middle-income countries (LMICs) have witnessed an increase in the prevalence of cardiometabolic risk factors such

as diabetes, hypertension and obesity in the adult population.[1–3] Overweight and obesity are not only metabolic outcomes but also serve as risk factors for other cardiometabolic diseases.[4] The Developmental Origins of Health and Disease hypothesis proposes that anatomical, physiological and metabolic adaptations in early life have long-lasting effects on the onset of adult cardiometabolic disease.[5]

Adaptations in body composition might have an important role in the programming of cardiometabolic disease *'through its own programming by early growth, and/or through being a mediator of the programming process'.*[6] A recent review concluded that associations between infant growth and long-term cardiometabolic outcomes might be mediated by body composition after estimates were attenuated when controlling for current body mass index (BMI).[7] Similarly, other systematic reviews and one meta-analysis showed that childhood obesity is a risk factor of cardiometabolic outcomes in adulthood. However, results were inconclusive in demonstrating whether or not this excessive weight gain in childhood was an independent factor or whether most of this effect was mediated through contemporary adult BMI.[8–10]

BMI in infancy, childhood and adolescence has a moderate to strong tracking into adult adiposity.[8 11] However, the discrimination between body compartments such as fat and lean mass, which have different metabolic functions, is more relevant to understand cardiometabolic risk than BMI alone.[12] The study of the role of growth across the life span on body composition has demonstrated similarities and differences when comparing body compartments. Overall studies have shown that birth weight is positively associated with lean mass later in life,[6] and that it is a stronger predictor of fat-free mass (FFM) than fat mass (FM).[13 14] Less consistency (positive, inverse and null findings) has been observed in studies that have examined the association between birth weight (low and high) with adiposity in childhood, adolescence and adulthood.[6 15 16] Additionally, studies in high-income countries (HICs) have shown that a rapid weight gain ($\geq 0.67\,SD$) in infancy (first 24 months after birth) is associated with higher obesity and FM in childhood and adulthood,[7 17] whereas findings in LMICs have shown that weight gain in infancy is associated with higher FM and FFM in adult life, being a stronger predictor of FFM.[6 7 17]

There is a paucity of studies about the long-term effects of growth across the life span on adult body composition. Available evidence has primarily examined associations between prenatal and postnatal growth (mainly in infancy) with body composition in childhood and adolescence. In a recent literature review of publications between 2003 and 2018, only 6 out of 39 studies examined associations between weight gain during infancy and body composition in adult participants from LMICs.[7] Less evidence is available about the role of linear growth (height) on adult body composition, and findings are mixed.[7] There is a need for investigating the independent long-term role of linear growth and relative weight

gain across the life course on body composition in young and mid-adulthood; stages in which the accumulation of cardiometabolic risk is higher.

Pooled analyses of the Consortium of Health-Oriented Research in Transitional Societies (COHORTS), a consortium of birth cohorts from five LMICs,[18] showed that higher weight at birth, in infancy and in mid-childhood were associated with a higher risk of adult overweight (ages 18–31 years).[13] These pooled analyses also showed that weight at birth and in infancy were stronger predictors of FFM than of FM in adult life, while weight gain in mid-childhood was a stronger predictor of FM than FFM.[13 14] We expand on these previous studies by (1) Analysing for the first time the period between childhood and adolescence, (2) Examining the long-term role of child growth on body composition in young and mid-adulthood (ages 22–46 years) given that in previous analyses some study participants were adolescents, and (3) Using body composition indices (height squared adjustment) to assess body compartments (weight, FM, and FFM) independent of linear size. Specifically, we now examine the associations of linear growth and relative weight at birth, infancy, childhood and adolescence with measures of body size and composition (BMI, fat mass index (FMI), fat-free mass index (FFMI), FM/FFM ratio and waist circumference) in young and mid-adulthood.

## METHODS
### Study design and participants
We conducted a secondary data analysis of COHORTS.[18] This consortium comprises six prospective birth cohorts in five study sites: (1) The 1982[19 20] and 1993 Pelotas Birth Cohorts in Brazil;[21] (2) The Institute of Nutrition of Central America and Panama Nutrition Trial and Longitudinal Study in Guatemala;[22] (3)The New Delhi Birth Cohort in India;[23] (4) The Cebu Longitudinal Health and Nutrition Survey in the Philippines;[24] and (5) The Birth to Twenty Plus Cohort in South Africa.[25] Table 1 presents a general description of each birth cohort.

We selected one analytical sample for each study site (figure 1). Of a total of 27 437 participants, we excluded participants with missing body composition in adulthood (n=16 029), implausible values of BMI $\leq 12\,kg/m^2$ or $\geq 60\,kg/m^2$ and waist circumference $\leq 51\,cm$ or $\geq 190\,cm$ (n=1), and pregnancy at the moment of the anthropometric assessment (n=135). We also excluded those individuals with missing anthropometric data at birth, childhood or adolescence (n=6967), implausible values of weight and height considering the WHO Z-score thresholds (n=168),[26 27] and pregnancy at the adolescence measurement (n=47).

## VARIABLE SPECIFICATION
### Outcomes: adult body composition
Study sites applied different standard methods to measure body composition in adult participants such

**Table 1** General characteristics of the six birth cohorts and body composition methods used at each study site

| Country (cohort) | Study design | Cohort enrolment | Age at enrolment | Initial sample | Analytical sample | Age and year at most recent follow-up | Body composition methods |
|---|---|---|---|---|---|---|---|
| Brazil (1982 Pelotas Birth Cohort) | Prospective cohort | 1982 | Birth | 5914 | 674 | 30 (2012) | BodPod and anthropometry |
| Brazil (1993 Pelotas Birth Cohort) | Prospective cohort | 1993 | Birth | 5249 | 827 | 22.5 (2015) | BodPod and anthropometry |
| Guatemala (INCAP Nutrition Trial and Longitudinal Study) | Community trial | 1969–1977 | Pregnancy to 7 years | 2392 | 163 | 45.5 (2015–2017) | Deuterium oxide dilution technique and anthropometry |
| India (NDBC) | Prospective cohort | 1969–1972 | Before pregnancy | 8181 | 681 | 46 (2016–2019) | Bioimpedance and anthropometry |
| The Philippines (CLHNS) | Prospective cohort | 1983–1984 | Birth | 3080 | 1197 | 34.5 (2018) | Bioimpedance and anthropometry |
| South Africa (Birth to Twenty Plus Cohort) | Prospective cohort | 1990 | Pregnancy | 3273 | 595 | 22 (2012) | DEXA (Dual-energy X-ray absorptiometry) and anthropometry |

INCAP, Institute of Nutrition of Central America and Panama; NDBC, New Delhi Birth Cohort; CLHNS, Cebu Longitudinal Health and Nutrition Survey; BodPod, air-displacement plethysmography.

as dual-energy X-ray absorptiometry, air-displacement plethysmography (BodPod), deuterium oxide dilution technique, bioimpedance[19 28–30] and anthropometry (table 1). Our outcomes measures are: (1) BMI (weight in kg/height in metres$^2$) as a proxy of body size; (2) FMI to represent the fat compartment adjusted for size (FM in kg/height in metres$^2$); (3) FFMI to represent the lean compartment adjusted for size (FFM in kg/height in metres$^2$); (4) FM/FMM ratio (a higher ratio indicates a greater increment in the FM component); and (5) The waist circumference (centimetres) as a proxy of abdominal adiposity. We transformed these outcomes measures into Z-scores (SD units) to compare effect sizes within strata of site and sex, and among the measures.

### Anthropometry at birth, infancy, childhood and adolescence

Birth weight was measured in all study sites, while birth length was available only in four of the six study sites (except Brazil 1982 and South Africa). In Brazil, anthropometric data at birth was measured at the hospitals by the research team (Brazil 1993) or by the hospital personnel (Brazil 1982) using paediatric scales.[31] In Guatemala, the study team measured weight within the first 48 hours after birth and length within the first 15 days

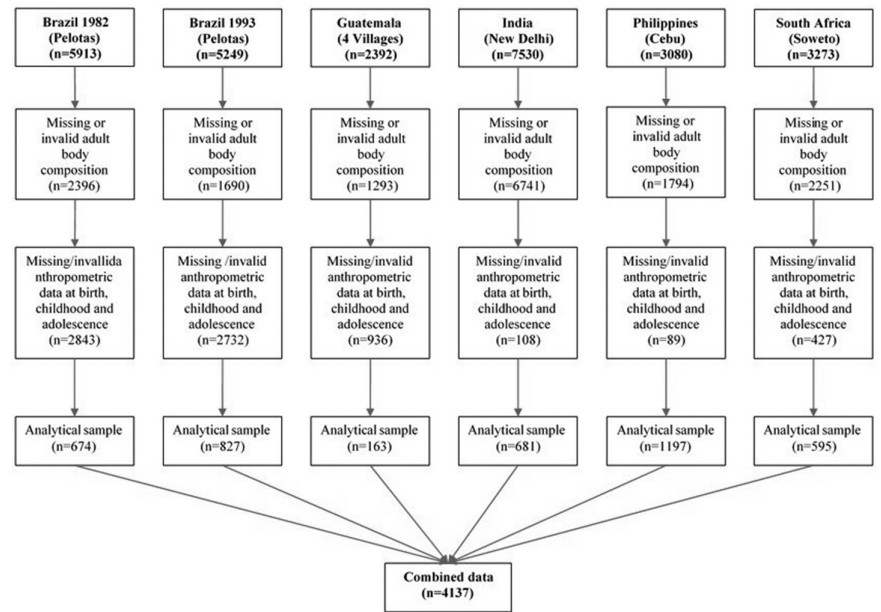

**Figure 1** Flow chart of analytical samples selection, Consortium of Health-Oriented Research in Transitional Societies (COHORTS).

after the delivery, using a beam balance and a measuring board.[32] In India, trained project staff measured birth and length within the first 72 hours of life using standard instruments.[33] In the Philippines, the 60% of the participants who were born at home were weighted by local birth attendants using hanging scales and the remaining participants were measured in hospitals, and birth length was measured within the first 6 days of life.[34] In South Africa, birth weight measured by hospital personnel was obtained from hospital records.

Trained research staff at each study site followed standard procedures to take anthropometric measures of length and/or height and weight in subsequent follow-ups (childhood and adolescence). Growth measures that represent linear growth (length/height) and mass accretion independent of linear size (relative weight) were measured in different periods at each study site. We identified four common or near ages where anthropometric data were available across all cohorts: (1) Birth, (2) Infancy (all study sites registered weight and length data at 24 months of age except Brazil 1993 that collected these data at 12 months), (3) Childhood (five study sites collected growth data at 48 months of age, but the Philippines collected data at 102 months), and (4) Adolescence (all study sites collected anthropometric data at 180 months of age). To be able to compare growth measures between study sites, we estimated sex-specific length/height-for-age and weight-for-age Z-scores (HAZ and WAZ, respectively) using the WHO growth standards.[35]

### Exposures: conditional growth (height and relative weight)

Repeated measures of linear growth and relative weight are strongly correlated. To eliminate this correlation, we estimated conditional size measures (standardised residuals) by regressing current HAZ on previous HAZ and WAZ measures, and regressing current WAZ on current HAZ and all prior HAZ and WAZ measures, within strata of site and sex. This approach allowed us to identify if children's growth deviates from their expected length/height or weight as estimated based on their own previous growth measurements and the growth of other children. We used birth weight as the anchor to estimate conditional growth measures (height and relative weight) in the main analysis of all cohorts. In separate models, we derived conditional growth measures using birth length as an anchor in the four study sites in which this variable was available. We estimated conditional height and conditional relative weight in four age intervals across all study sites: (1) Prenatal (from conception to birth captured though birth weight and birth length); (2) Infancy (birth to 12 months in Brazil 1993, and birth to 24 months in the other study sites); (3) Childhood (12–48 months in Brazil 1993, 24–48 months in Brazil 1982, Guatemala, India, and South Africa, and 24–102 months in the Philippines); and (4) Adolescence (48–180 months in five study sites and 102–180 months in the Philippines).

*Covariates:* We identified potential confounders available at each study site such as birth characteristics (gestational age, birth order), income and/or wealth index of child's household at birth, maternal characteristics (height, age at first childbirth, schooling attainment, marital status), paternal characteristics (age, schooling attainment), reproductive factors (age at menarche) and age in adulthood (online supplemental table 1).

## STATISTICAL ANALYSES

We used R V.4.0 for all statistical analyses. Statistical significance was established at a value of $p < 0.05$.

### Site-specific regression models

We used multiple linear regressions with standardised coefficients to estimate the associations between growth patterns at the four age intervals and adult body composition since all outcomes were continuous and without major violations of the normal distribution (graphical examination). We built site-specific models stratified by sex given the biological differences in body composition.[36] We ran two sets of models: (1) Unadjusted models for all study sites except Guatemala, in which we minimally adjust for birth year and intervention group considering the initial study design of this cohort;[22] (2) Fully adjusted models controlling for potential confounders available at each study site (online supplemental table 1). Additionally, we adjusted for adult height in waist circumference models, for age at menarche and teenage childbearing in models of women, and for maternal skin colour in both Brazil cohorts as one proxy of socioeconomic status. In Brazil 1993 models, we used survey procedures given that this cohort collected data in subsamples (all children with low weight at birth and a random sample of other children) at 12 months and 48 months of age.[31] After a visual examination of effect sizes from both sets of models at each study site, we observed slight changes in magnitude but not in direction (results not presented). We present the fully adjusted models to account for the confounding effect of this group of covariates.

### Pooling effect sizes

The cohorts come from different study sites and data were collected in different periods, resulting in heterogeneity among site-specific estimates. We used a random-effects meta-analysis approach to pool effect sizes to account for this underlying heterogeneity. Given that our outcomes were continuous, we used the restricted maximum likelihood estimator to calculate the between-study heterogeneity (variance of the distribution of true effect sizes).[37] We also applied Knapp-Hartung adjustments in the estimation of CIs of pooled effects. We estimated Cochran's Q with its corresponding p value to test whether there was heterogeneity between study sites, and calculated the $I^2$ statistic to quantify the magnitude of between-study heterogeneity. As a rule of thumb an $I^2$ equal to 25% is interpreted as low heterogeneity, 50% as moderate heterogeneity and 75% as substantial heterogeneity.[37] We used the Metafor package in R to run these analyses.

## Multiple imputation

To deal with missing data, we built sex-specific multiple imputation models at each study site. Missing data patterns are presented in online supplemental table 1. We assumed that the underlying missing pattern was missing at random (MAR). In brief, we used the R package Multivariate Imputation by Chained Equations to generate different multiple imputed data sets with a maximum of 50 iterations. We estimated the number of imputations based on the highest percentage of missingness within strata of site and sex. In our imputation models, we included all covariates used in the main analyses and the outcome variables. We used the 'with and pool' functions to run the main analyses in each imputed data set and to combine the estimates into a single result with its corresponding SEs and CIs.

## Patient and public involvement

Patients and/or the public were not involved in the design, or conduct, or reporting, or dissemination plans of our research.

## RESULTS

We analysed data from a combined analytical sample of 4137 participants with complete and valid information of body composition in adulthood and anthropometry across the life course (figure 1). The comparison of general characteristics between participants included in the final analytical samples versus those excluded only showed significant differences in a couple of variables in selected cohorts (online supplemental table 2). General characteristics of the six cohorts, and anthropometric characteristics at birth and adulthood of men and women at each study site are presented in tables 1 and 2, respectively. Participants' average adult age ranged from 22 years in Brazil 1993 and South Africa to 45–46 years in Guatemala and India. India had the lowest mean birth weight compared with other study sites. The proportion of excess weight and adiposity were correlated with the age of the cohorts, with the oldest cohorts having the greatest proportion of BMI $\geq 25 \, \text{kg/m}^2$ and the highest FMI. Among men, the proportion of excess weight ranged from 16.0% in South Africa to 73.2% in India, and among women it ranged from 44.5% in Brazil 1993 to 85.4% in Guatemala. Participants from Guatemala, India and Brazil 1982 had the highest FMI (women above $10 \, \text{kg/m}^2$ and men above $7 \, \text{kg/m}^2$). FFMI ranged from 15.5 $\text{kg/m}^2$ to 20.2 $\text{kg/m}^2$ in men, and from 13.7 $\text{kg/m}^2$ to 17.7 $\text{kg/m}^2$ in women. Other characteristics (at birth, childhood and parental) of study participants at each study site are presented in online supplemental table 3.

## Conditional relative weight and adult body composition

Table 3 shows the pooled adjusted associations of conditional growth with adult FMI and FFMI. Birth weight and conditional relative weight in infancy, childhood and adolescence were positively associated with FMI and FFMI

in adulthood. A higher birth weight was a predictor of higher FMI (men: 0.07 (95% CI 0.01 to 0.14); women: 0.09 (95% CI 0.01 to 0.17)) and FFMI (men: 0.11 (95% CI 0.05 to 0.18); women: 0.12 (95% CI 0.05 to 0.19)). An increment of 1 SD in conditional relative weight in infancy was associated with an increment of 0.2–0.3 SD in adult FMI, while for every SD increment in conditional weight in childhood or adolescence there was a 0.4–0.5 SD increase in FMI in adulthood. Similarly, the associations of relative weight gain with adult FFMI were positive and stronger as boys and girls grew older. Increments in conditional relative weight by 1 SD in infancy, childhood and adolescence were associated with higher FFMI by 0.3 SD, 0.4–0.5 SD and 0.4 SD units in adult life, respectively. Table 4 shows the pooled adjusted associations of conditional growth with adult FM/FFM ratio. Birth weight was not significantly associated with FM/FFM ratio in adult life. Conditional relative weight in infancy, childhood and adolescence were associated with a higher FM/FMM ratio with increments ranging from 0.17 to 0.43 for every SD in conditional weight.

The between-study heterogeneity assessment showed low or moderate heterogeneity across almost all associations of conditional growth with FMI and FM/FFM, except conditional relative weight in childhood among women ($I^2$=83.9%) and conditional relative weight in adolescence among men ($I^2$=78.0%) (tables 3 and 4). The between-study heterogeneity assessment of FFMI models showed substantial variability in estimates of conditional relative weight in childhood and adolescence among women and men, and conditional height in childhood and adolescence among women ($I^2 \geq 75\%$) (table 3).

Site-specific estimates were consistent in magnitude and direction. Birth weight was positively associated with adult FMI in men and women from India and the Philippines, and in women from Brazil 1993. Conditional relative weights in infancy, childhood and adolescence were positively associated with adult FMI among men and women across all cohorts (online supplemental table 4). Site-specific analyses showed that birth weight was a predictor of a higher adult FFMI among men and women from four out of the six cohorts (Brazil 1982, India, the Philippines and South Africa), and among women from Brazil 1993 and Guatemala. Across all cohorts but Guatemala, a higher relative weight gain in childhood was significantly associated with a higher FFMI. Conditional relative weight in adolescence was a strong and significant predictor of a higher FFMI in adult life among all study sites and both sexes (online supplemental table 5). Birth weight was positively and significantly associated with FM/FFM among men and women from India and the Philippines, and women from Brazil 1993. Conditional relative weights in infancy, childhood and adolescence were positively associated with FM/FFM across all study sites with a few exceptions (online supplemental table 6).

Table 5 shows the pooled adjusted associations of conditional growth with adult BMI and waist circumference. Birth weight and all conditional relative weight

**Table 2** Anthropometric characteristics of participants included in the analysis, stratified by study site and sex*

| | Brazil 1982 | | Brazil 1993 | | Guatemala | | India | | The Philippines | | South Africa | |
|---|---|---|---|---|---|---|---|---|---|---|---|---|
| | Men (n=344) | Women (n=330) | Men (n=375) | Women (n=452) | Men (n=67) | Women (n=96) | Men (n=426) | Women (n=255) | Men (n=656) | Women (n=541) | Men (n=287) | Women (n=308) |
| Birth length (cm) | NA | NA | 49.4 (2.3) | 48.4 (2.1) | 49.6 (2.6) | 48.5 (2.2) | 48.7 (2.0) | 47.9 (2.1) | 49.3 (2.0) | 48.7 (2.0) | NA | NA |
| Birth weight (kg) | 3.3 (0.6) | 3.2 (0.5) | 3.3 (0.5) | 3.1 (0.5) | 3.1 (0.5) | 3.0 (0.5) | 2.9 (0.4) | 2.7 (0.4) | 3.0 (0.4) | 3.0 (0.4) | 3.1 (0.5) | 3.0 (0.5) |
| Adult height (cm) | 174.9 (7.1) | 161.9 (6.2) | 174.8 (7.6) | 160.8 (6.6) | 164.7 (5.7) | 152.0 (4.8) | 169.9 (6.8) | 154.5 (5.5) | 162.8 (5.6) | 150.9 (5.5) | 171.5 (6.2) | 159.7 (6.4) |
| Adult weight (kg) | 82.8 (16.3) | 70.1 (16.6) | 75.8 (16.1) | 66.4 (15.6) | 71.4 (14.3) | 69.4 (11.8) | 80.6 (15.0) | 70.0 (12.5) | 66.1 (12.8) | 57.3 (11.9) | 62.9 (11.3) | 64.6 (14.9) |
| Adult body mass index (BMI) (kg/m²†) | 27.0 (4.8) | 26.7 (6.1) | 24.6 (4.7) | 25.5 (5.9) | 26.3 (4.8) | 30.0 (4.7) | 27.9 (4.7) | 29.3 (4.8) | 24.9 (4.4) | 25.2 (5.0) | 21.4 (3.5) | 25.3 (5.6) |
| Adult excess weight† | 215 (62.5%) | 171 (51.8%) | 160 (40.4%) | 195 (44.5%) | 42 (62.7%) | 82 (85.4%) | 312 (73.2%) | 208 (81.6%) | 297 (45.3%) | 256 (47.3%) | 46 (16.0%) | 142 (46.1%) |
| Adult waist circumference (cm) | 89.9 (11.5) | 80.8 (11.7) | 82.2 (10.8) | 77.3 (11.6) | 93.7 (10.8) | 102.8 (10.7) | 100.6 (12.0) | 91.8 (10.5) | 82.1 (10.8) | 80.8 (11.5) | 75.8 (8.6) | 82.4 (13.5) |
| Adult fat mass index (FMI, kg/m²†) | 7.1 (3.6) | 10.5 (4.7) | 5.3 (3.7) | 9.6 (4.5) | 7.7 (2.8) | 12.7 (3.1) | 7.7 (2.8) | 11.6 (3.4) | 5.5 (2.5) | 9.0 (3.5) | 4.0 (2.1) | 9.6 (3.9) |
| Adult fat-free mass index (FFMI, kg/m²†) | 19.9 (2.1) | 16.2 (1.9) | 19.2 (1.9) | 15.8 (1.9) | 18.5 (2.6) | 17.3 (2.2) | 20.2 (2.2) | 17.7 (1.7) | 19.3 (2.0) | 16.1 (1.4) | 15.5 (1.8) | 13.7 (1.9) |
| Adult fat mass/fat-free mass ratio | 0.4 (0.2) | 0.6 (0.2) | 0.3 (0.2) | 0.6 (0.2) | 0.4 (0.1) | 0.7 (0.1) | 0.4 (0.1) | 0.6 (0.2) | 0.3 (0.1) | 0.5 (0.2) | 0.3 (0.1) | 0.7 (0.2) |

*Values are means (SD) or n (percentages). Sample sizes (n) refer to participants with non-missing data for outcomes (adult body composition), and non-missing data for exposures (growth measures).
†Excess weight defined as a BMI ≥25 kg/m²
NA, not available data.

**Table 3** Pooled adjusted associations between weight at birth, conditional growth (height and relative weight) in infancy, childhood, and adolescence with adult fat mass index (FMI) and fat-free mass index (FFMI), stratified by sex*

| | FMI (SD units) | | | | | | FFMI (SD units) | | | | | |
|---|---|---|---|---|---|---|---|---|---|---|---|---|
| | Men (n=2155) | | | Women (n=1982) | | | Men (n=2155) | | | Women (n=1982) | | |
| | β (95% CI) | I² Statistic (%)‡ | Cochran's Q test (P value) | β (95% CI) | I² Statistic (%) | Cochran's Q test (P value) | β (95% CI) | I² Statistic (%) | Cochran's Q test (P value) | β (95% CI) | I² Statistic (%) | Cochran's Q test (P value) |
| **Conditional growth, Z-scores†** | | | | | | | | | | | | |
| Birth weight | 0.07* (0.01 to 0.14) | 42.4 | 0.147 | 0.09* (0.01 to 0.17) | 57.4 | 0.044 | 0.11* (0.05 to 0.18) | 47.0 | 0.070 | 0.12* (0.05 to 0.19) | 9.0 | 0.040 |
| Conditional relative weight in infancy | 0.23* (0.18 to 0.27) | 0.0 | 0.868 | 0.26* (0.19 to 0.33) | 0.1 | 0.498 | 0.30* (0.23 to 0.37) | 24.5 | 0.297 | 0.30* (0.20 to 0.39) | 50.3 | 0.104 |
| Conditional relative weight in early/mid-childhood | 0.41* (0.26 to 0.55) | 61.9 | 0.020 | 0.46* (0.21 to 0.72) | 83.9 | <0.001 | 0.50* (0.34 to 0.66) | 75.9 | 0.001 | 0.42* (0.17 to 0.67) | 92.2 | <0.001 |
| Conditional relative weight in adolescence | 0.39* (0.27 to 0.50) | 77.4 | 0.001 | 0.53* (0.44 to 0.61) | 50.0 | 0.101 | 0.43* (0.32 to 0.55) | 81.9 | <0.001 | 0.39* (0.13 to 0.66) | 95.1 | <0.001 |
| Conditional height in infancy | 0.08* (0.03 to 0.14) | 21.2 | 0.260 | 0.11* (0.07 to 0.16) | 12.9 | 0.558 | 0.09 (0.00 to 0.19) | 69.2 | 0.003 | 0.09 (−0.01 to 0.20) | 78.5 | 0.001 |
| Conditional height in early/mid-childhood | 0.08 (−0.02 to 0.18) | 42.1 | 0.156 | 0.14* (0.04 to 0.24) | 31.2 | 0.285 | 0.06 (−0.06 to 0.17) | 58.4 | 0.039 | 0.12 (−0.14 to 0.38) | 90.0 | <0.001 |
| Conditional height in adolescence | 0.02 (−0.08 to 0.12) | 16.1 | 0.192 | 0.01 (−0.13 to 0.15) | 56.1 | 0.060 | 0.00 (−0.08 to 0.08) | 17.6 | 0.314 | 0.06 (−0.16 to 0.28) | 89.3 | <0.001 |

*Values are standardised linear regression coefficients (βs and 95% CIs). In adjusted analyses we controlled for birth characteristics (gestational age, birth order), maternal characteristics (height, age at childbirth, schooling, marital status), paternal schooling, income and/or wealth index of child's household at birth, and age in adulthood. In women' models, we adjusted for age of menarche and teenage childbearing. Additionally, we controlled for maternal skin colour in both Brazil cohorts and adjusted for birth year and intervention group (village fixed effects) in Guatemala analysis. * p<0.05.
†We used relative birth weight as the anchor in all conditional models given that this measure was available across all study sites. In the supplementary section, we present similar models using birth length as the anchor in the sites that collected this measure.
‡As a rule of thumb I²=25% is interpreted as low heterogeneity, 50% moderate heterogeneity and 75% substantial heterogeneity.

**Table 4** Pooled adjusted associations between weight at birth, conditional growth (height and relative weight) in infancy, childhood and adolescence with adult fat mass/fat-free mass ratio (FM/FFM), stratified by sex*

| | FM/FFM (SD units) | | | | | |
| | Men (n=2155) | | | Women (n=1982) | | |
| | β (95% CI) | I²† Statistic (%)‡ | Cochran's Q test (P value) | β (95% CI) | I²† Statistic (%) | Cochran's Q test (P value) |
|---|---|---|---|---|---|---|
| **Conditional growth, Z-scores†** | | | | | | |
| Birth weight | 0.06 (−0.03 to 0.14) | 64.3 | 0.032 | 0.06 (−0.03 to 0.15) | 68.0 | 0.013 |
| Conditional relative weight in infancy | 0.17* (0.12 to 0.23) | 0.0 | 0.685 | 0.20* (0.09 to 0.31) | 32.7 | 0.108 |
| Conditional relative weight in early/mid-childhood | 0.31* (0.16 to 0.47) | 65.9 | 0.009 | 0.37* (0.07 to 0.66) | 87.5 | <0.001 |
| Conditional relative weight in adolescence | 0.31* (0.19 to 0.43) | 78.0 | 0.002 | 0.43* (0.39 to 0.48) | 0.0 | 0.803 |
| Conditional height in infancy | 0.07* (0.02 to 0.12) | 25.0 | 0.336 | 0.10* (0.03 to 0.16) | 37.6 | 0.181 |
| Conditional height in early/mid-childhood | 0.08 (0.00 to 0.17) | 28.7 | 0.313 | 0.16* (0.08 to 0.25) | 15.1 | 0.477 |
| Conditional height in adolescence | 0.02 (−0.08 to 0.13) | 27.2 | 0.124 | −0.01 (−0.19 to 0.17) | 72.0 | 0.002 |

*Values are standardised linear regression coefficients (βs and 95% CIs). In adjusted analyses we controlled for birth characteristics (gestational age, birth order), maternal characteristics (height, age at childbirth, schooling, marital status), paternal schooling, income and/or wealth index of child's household at birth, and age in adulthood. In women' models, we adjusted for age of menarche and teenage childbearing. Additionally, we controlled for maternal skin colour in both Brazil cohorts and adjusted for birth year and intervention group (village fixed effects) in the Guatemala analysis. * p<0.05.
†We used relative birth weight as the anchor in all conditional models given that this measure was available across all study sites. In the supplementary section, we present similar models using birth length as the anchor in the sites that collected this measure.
‡As a rule of thumb I²=25% is interpreted as low heterogeneity, 50% moderate heterogeneity and 75% substantial heterogeneity

measures were positively associated with BMI and waist circumference among adult men and women. Effect sizes at birth were small but significant predictors of BMI and waist circumference. Associations strengthened with age at measurement; for example, men's BMI increased by 0.29 (95% CI 0.24 to 0.35) SD units, 0.50 (95% CI 0.37 to 0.64) SD units and 0.47 (95% CI 0.35 to 0.60) SD units per each SD in conditional relative weight in infancy, childhood and adolescence, respectively. Similar patterns were observed in women. Relative weight gains in infancy and childhood were associated with increments in waist circumference ranging from 0.25 SD to 0.26 SD and 0.40–0.43 SD among adult men and women, respectively. Increments in relative weight during adolescence predicted increments in waist circumference by 0.39 (95% CI 0.27 to 0.52) SD units in men and 0.49 (95% CI 0.42 to 0.56) SD units in women. The between-study heterogeneity was low or moderate for most of the associations, except some associations of conditional relative weight in childhood and adolescence (heterogeneities above 75%) (table 5).

Site-specific analyses of BMI and waist circumference showed similar effect sizes in direction and magnitude as pooled estimates. Relative weight gains in infancy, childhood and adolescence were predictors of higher BMI and

waist circumference in adulthood across study sites, with a few exceptions (online supplemental tables 7 and 8).

### Conditional height and adult body composition
Conditional height in infancy was positively associated with FMI and FM/FFM ratio in adult life, but effect sizes were small (around 0.1) (tables 3 and 4). None of the conditional height variables were significantly associated with adult FFMI (table 3). A high conditional height in infancy was a significant but weak predictor of higher adult BMI among men and women, and waist circumference among women (increments of 0.10–0.12 SD units) (table 5). Conditional height in childhood was positively but weakly associated with women's FMI, FM/FFM ratio and waist circumference (tables 3–5).

We used conditionals derived from birth length as the anchor in models in four of the study sites. In these pooled analyses, we found that birth length was a weak but statistically significant predictor of higher FFMI and BMI in adult women (online supplemental tables 9 and 10). Other effect sizes from these models were consistent in direction and magnitude with effect sizes observed in the main models (birth weight as anchor). Site-specific

**Table 5** Pooled adjusted associations between weight at birth, conditional growth (height and relative weight) in infancy, childhood and adolescence with adult body mass index (BMI) and waist circumference, stratified by sex*

| | BMI (SD units) | | | | | | Waist circumference (SD units) | | | | | |
| | Men (n=2155) | | | Women (n=1982) | | | Men (n=2155) | | | Women (n=1982) | | |
| | β (95% CI) | I²† Statistic (%)‡ | Cochran's Q test (P value) | β (95% CI) | I²† Statistic (%) | Cochran's Q test (P value) | β (95% CI) | I²† Statistic (%) | Cochran's Q test (P value) | β (95% CI) | I²† Statistic (%) | Cochran's Q test (P value) |
|---|---|---|---|---|---|---|---|---|---|---|---|---|
| **Conditional growth, Z-scores†** | | | | | | | | | | | | |
| Birth weight | 0.10* (0.05 to 0.14) | 13.7 | 0.454 | 0.12* (0.03 to 0.20) | 47.0 | 0.056 | 0.08* (0.03 to 0.12) | 0.0 | 0.573 | 0.09* (0.01 to 0.16) | 0.1 | 0.260 |
| Conditional relative weight in infancy | 0.29* (0.24 to 0.35) | 0.0 | 0.602 | 0.30* (0.24 to 0.36) | 0.0 | 0.634 | 0.26* (0.18 to 0.33) | 17.9 | 0.345 | 0.25* (0.19 to 0.32) | 0.0 | 0.638 |
| Conditional relative weight in early/mid-childhood | 0.50* (0.37 to 0.64) | 58.6 | 0.044 | 0.53* (0.31 to 0.75) | 78.4 | 0.001 | 0.43* (0.35 to 0.52) | 0.0 | 0.551 | 0.40* (0.16 to 0.63) | 79.3 | 0.002 |
| Conditional relative weight in adolescence | 0.47* (0.35 to 0.60) | 85.5 | <0.001 | 0.58* (0.47 to 0.69) | 69.4 | 0.006 | 0.39* (0.27 to 0.52) | 79.9 | <0.001 | 0.49* (0.42 to 0.56) | 30.0 | 0.285 |
| Conditional height in infancy | 0.10* (0.03 to 0.16) | 36.5 | 0.097 | 0.12* (0.08 to 0.16) | 0.0 | 0.736 | 0.12* (0.03 to 0.21) | 48.2 | 0.089 | 0.08 (-0.05 to 0.20) | 19.8 | 0.216 |
| Conditional height in early/mid-childhood | 0.07 (-0.04 to 0.19) | 58.4 | 0.043 | 0.12 (-0.01 to 0.25) | 53.2 | 0.066 | 0.11 (-0.04 to 0.26) | 57.3 | 0.04 | 0.14* (0.06 to 0.23) | 0.0 | 0.866 |
| Conditional height in adolescence | 0.01 (-0.07 to 0.09) | 0.0 | 0.307 | 0.01 (-0.1 to 0.13) | 37.3 | 0.225 | 0.08 (0.00 to 0.16) | 0.0 | 0.596 | -0.04 (-0.26 to 0.18) | 41.7 | 0.161 |

*Values are standardised linear regression coefficients (βs and 95% CIs). In adjusted analyses we controlled for birth characteristics (gestational age, birth order), maternal characteristics (height, age at childbirth, schooling, marital status), paternal schooling, income and/or wealth index of child's household. In women† models, we adjusted for age of menarche and teenage childbearing. Additionally, we controlled for maternal skin colour in both Brazilian cohorts and adjusted for birth year and intervention group (village fixed effects) in the Guatemala analysis. * p<0.05.

†We used relative birth weight as the anchor in all conditional models given that this measure was available across all study sites. In the supplementary section, we present similar models using birth length as the anchor in the sites that collected this measure.

‡As a rule of thumb: I²=25% is interpreted as low heterogeneity; 50% moderate heterogeneity and 75% substantial heterogeneity.

analyses using birth length as anchor are presented in online supplemental tables 11–14.

Body size and composition indices should control for allometric scaling (differences in body compartments given differences in linear size) and should be independent of their denominator (height).[38 39] Thus, to assess the statistical validity of body size and composition indices, we examined the correlation between these indices with adult height. We observed that the three indices (BMI, FMI and FFMI) were not significantly correlated with height across all study sites and sexes (correlation coefficients ranged from −0.15 to 0.10). The correlation coefficients of waist circumference with adult height were significant but weak (0.15–0.26) (online supplemental table 15).

## DISCUSSION

In a pooled analysis of six prospective birth cohorts from five LMICs, we found that a higher birth weight predicted higher adult FFMI, FMI, BMI and waist circumference, but not adiposity (FM/FFM). Birth length was a positive predictor of FFMI and BMI, only among adult women. Higher relative weight gains in infancy, childhood and adolescence were associated with higher FFMI, FMI, FM/FFM ratio, BMI and waist circumference in adult life. Associations strengthened with age, thus, effect sizes were the strongest in childhood and adolescence, almost twice the magnitude of effect sizes in infancy, and four to five times the magnitude of birth estimates. Our findings also showed that a higher conditional height in infancy was a significant but weak predictor of higher FMI and BMI among adult men and women, and higher waist circumference among women. A higher conditional height in childhood was positively but weakly associated with women's FMI, FM/FFM ratio and waist circumference.

This updated analysis confirms previous results from COHORTS[13 14] and similar site-specific studies[28 33 40] showing that birth weight is a predictor of higher adult FFMI and FMI. Thus, our findings showed the independent and long-lasting role of prenatal growth (weight gain mainly) on body composition and size in young and middle-age adulthood. There are differences in effect sizes when comparing our estimates of FMI and FFMI with previous COHORTS' analysis because we adjusted by adult height, used a different approach to combine site-specific estimates, participants were older, and possibly due to the decreased association of birth weight with adult measures across time. The positive and significant associations between prenatal and postnatal relative weight gains with all outcomes in young and mid-adulthood suggest that relative weight is a predictor of overall body size than body composition. A recent review of literature highlighted a similar conclusion in relation to rapid weight gain in infancy (0–2 years of age) with body size in childhood, adolescence and adulthood.[7] We expand on that conclusion by showing that excessive relative weight gain beyond 2 years of age is also a

predictor of overall size in young and middle-age adulthood. Additionally, we infer that a higher FFMI reflects the accretion in FFM as a body response (expansion of bone and muscle mass) to support higher weight,[41 42] since on average COHORTS participants had overweight or obesity in adulthood. This inference is also supported by the finding that conditional relative weight in infancy, childhood and adolescence were associated with a greater FM/FFM ratio, indicating that the increments in FM were faster than the increments in FMM. Thus, higher weight gains during these age intervals might be associated with a greater risk of adiposity in adult life.

Compared with previous combined analyses of COHORTS, we examined for the first time the interval from childhood to adolescence. Our findings showed that relative weight gain in adolescence was associated with higher adult adiposity (FMI, FM/FFM ratio and waist circumference) in consistency with earlier studies in LMICs.[13 14 28] Studies in HICs, like an analysis of the Fels Longitudinal Study showed similar trends. This study found that timing and velocity of BMI in childhood, adolescence and postadolescence were predictors of higher BMI and adiposity in adulthood (ages 35–45 years), adolescence and postadolescence being the strongest predictors of adult adiposity.[43] Whereas we observed similar effect sizes in childhood and adolescence suggesting that both periods are equally important for predicting the risk of higher adiposity in adult life. In a deeper understanding of the predicting role of growth on adult adiposity, recent analyses of Brazil 1982[44] and South Africa[28] cohorts found that conditional relative weight gains across different stages of the life course were significant predictors of both visceral and subcutaneous fat at ages 30 years and 22 years, respectively.[44]

A combination of biological, environmental, behavioural and sociodemographic factors is associated with weight gain and adiposity in childhood and adolescence that may not only persist later in life but might undermine the positive role of a healthy birth weight and linear growth early in life. Some of these factors are the medium to strong tracking of child and adolescent BMI into adult adiposity,[8 11] prenatal and postnatal factors,[45] reproductive characteristics of women (earlier age at menarche, short periods between age at menarche and age at first pregnancy),[46] and changes in lifestyle patterns (diets of obesogenic environments, a decline in physical activity, and an increment in sedentary activities) as consequence of the nutrition transition and transformations of the food systems, home and work environments in LMICs during the last decades.[47]

Fewer studies have examined the role of linear growth on later body composition and results are mixed.[7] Our results showed that linear growth in infancy was a significant but weak predictor of adiposity in adult life (FMI and BMI in men and women, and waist circumference in women). These findings are also consistent with previous reports of COHORTS[13] and South Africa birth cohort,[28] but our estimates were smaller given the adjustment by

adult height. Our findings showed that linear growth was not a predictor of adult FFMI, which suggests that linear growth is a predictor of adult height rather than FFM. In contrast, an earlier study in South Africa showed that linear growth across five age intervals (from 0 to 22 years of age) predicted a higher FFM in young adulthood,[28] and an analysis conducted in Guatemala before the start of the obesity epidemic found that linear growth was associated with higher FFM in adolescence.[48]

Linear growth is determined by the hormonal regulation of the growth hormone (GH) secreted by the pituitary and the insulin-like growth factor 1 (IGF-1), an intermediary hormone between the GH and target tissues.[49] Based on human models, growth during the first year after birth is described as an extension of fetal growth,[50] and linear size in infancy is influenced prominently by IGF-1 that is regulated mainly by nutritional factors than by the GH.[51 52] Thus, IGF-1 might be the hormonal link underlying the association between faster linear growth in infancy and adult adiposity given its metabolic functions (anabolism) and role in promoting linear growth.[49] Studies of British cohorts have shown that infants (0–2 years) with a faster postnatal growth (catch-up growth) in weight or height, had higher levels of IGF-1 at age 5 years, independent of birth levels.[53] IGF-1 levels in childhood were positively correlated with weight, height and body composition in childhood.[53] Besides, children who are taller tend to be heavier, and rapid weight gains and catch-up growth in postnatal years have been associated with higher adiposity and risk of obesity in childhood and adult life.[7] In turn, animal models and clinical trials have demonstrated that excess weight causes a downregulation of the GH/IGF-1 axis leading to higher adiposity in adulthood.[54]

Thus, increased growth in infancy, including a faster linear growth, might induce the long-lasting metabolic programming of the GH/IGF-1 axis and a subsequent increased risk of adiposity. However, the evidence on the role of linear growth on obesity risk and higher adiposity later in life is limited and inconclusive.[7] For example, a recent study of a British cohort showed that infant linear growth (ages 0–3 months) modestly improved prediction models of childhood adiposity, but the estimates were small and with a negative trend.[55] While, our findings showed that faster linear growth in infancy (0-24 months) has a weak but significant association with higher adult adiposity, a positive trend that contrasts with findings in children.

Overall, larger weight gains across the life course were associated with larger body size and adiposity in adult life among men and women. However, among women, estimates for size in adolescence tended to be stronger than those for childhood, in contrast to the men, in whom childhood estimated tended to be stronger. Regarding conditional height, a faster liner growth in infancy was positively but weakly associated with higher adult adiposity among both men and women, and faster linear growth in childhood was positively associated with higher adiposity among adult women but not in men.

Sex-based dimorphism in human growth patterns, body size and composition since the prenatal period, with pronounced differences in certain age intervals is well recognised.[56] In infancy, growth patterns are similar among boys and girls, a period characterised by a deceleration of linear growth and fat accretion after the first year of birth.[57] However, in childhood, after the adiposity rebound (around age 6 years), there is acceleration in linear growth and weight gain, accretion that is mainly due to gains in lean mass among boys and gains in FM among girls.[57] In adolescence, women reach puberty earlier and growth stops at a younger age compared with men. Thus, men have a longer prepubertal period for growth and in adolescence they experience a greater accretion in lean mass than FM, whereas weight gain in women is characterised by a greater accretion in FM and their growth velocity is slower than men.[50 57] These sex differences remain until adult life, where men and women on average have different body stature, composition, shape and fat distribution. These changes and sex differences in body size and composition are the result of hormonal regulations, mainly by the action of the GH/IGF-1 axis in infancy and childhood, plus the effect of steroid hormones in puberty.[50 57] Thus, sex dimorphisms in growth patterns and body composition might explain the differences in the effect sizes and trends among men and women that we observed. Public health recommendations and interventions should pay special attention to these differences in growth and body composition among boys and girls mainly in childhood and adolescence, to prevent the excess of adiposity and high prevalence of obesity in adult life (higher among women), since early stages of life.

Some limitations of our analysis include: (1) Attrition as one source of selection bias given the long-lasting follow-up of these birth cohorts; (2) Residual confounding since we could not control for covariates such as diet, reproductive and life style factors; (3) Missing data of covariates (ranges between 0.1%–41%), for which we assumed MAR and applied multiple imputation techniques; and (4) As expected there was substantial between-study heterogeneity, mostly in associations of conditional relative weight in childhood and adolescence. Some reasons that might explain the observed heterogeneity, beyond the baseline differences in places and contexts, include: (a) The age intervals definition was guided by data availability, as not all children were measured at the same intervals; (b) Differences in sample sizes across study sites given that we had to restrict the analysis to participants with complete anthropometric data in the four age intervals to be able to estimate conditional growth measures; (c) Differences in the techniques and instruments to measure body composition which might be a source of information bias (non-differential misclassification). In this regard, our research question intended to study the absolute increase in body composition not the relative estimate, and we assumed that all techniques, independent of the method and units used, ranked people in the same way. Additionally, we

transformed all body composition measurements into SD units in each cohort and estimated standardised βs to be able to compare estimates between sites. To minimise the previous limitations, we used the random meta-analysis approach that not only considers the underlying heterogeneity when pooling estimates but also allows us to control for differences in sample sizes across study sites.

Among the strengths of our analyses, we highlight the prospective and long follow-up of these six birth cohorts in LMICs. We used large sample sizes (except Guatemala), and analysed two compartments of body composition (beyond BMI) that were measured using gold standards and/or validated techniques. We studied the independent association of linear growth and weight gain with body composition through the use of conditional growth measures, which allowed us to break the high correlation between weight and height and to study the independent role of linear growth and relative weight gain in specific age intervals. We selected the best model at each study site, fully adjusted by multiple confounders, and applied a random-effects meta-analysis that by default considers the heterogeneity between study sites.

We are cautious with the generalizability of our results since we used selected birth cohorts which design was not intended to be representative of countries were information was collected, and given the social, demographic, economic, political and cultural differences in contexts and periods in which each cohort started. However, the consistency in magnitude and direction of effect sizes across study sites (six birth cohorts from five different cities around the world), indicates that our findings are robust and suggests that the long-term role of growth on body composition might apply to individuals from other LMICs with similar characteristics.

In summary, our findings suggest that: (1) Growth in the first 1000 days of life has a small and long-lasting association with adult body size and composition; (2) Childhood and adolescence exhibit the strongest associations between relative weight gain with body size and composition in adulthood; and (3) A faster linear growth in infancy (girls and boys) and childhood (girls) predicts increased adiposity in adult life. Historical efforts to reduce growth faltering in LMICs have been successful, with little evidence to date of increases in BMI for age.[30 58 59] However, once nutritional and other environmental constraints on growth are removed, further nutritional inputs beyond individual requirements are likely to increase adiposity. Thus, public health interventions might focus on promoting adequate linear growth in early life and a healthy weight gain throughout the life course.

## Author affiliations
[1]Hubert Department of Global Health, Rollins School of Public Health, Emory University, Atlanta, Georgia, USA
[2]Carolina Population Center, University of North Carolina at Chapel Hill, Chapel Hill, North Carolina, USA
[3]INCAP Research Center for the Prevention of Chronic Diseases (CIIPEC), Institute of Nutrition of Central America and Panama (INCAP), Guatemala City, Guatemala
[4]Department of Pediatrics, Safdarjang Hospital and Vardhman Mahavir Medical College, New Delhi, India
[5]Office of Population Studies Foundation, University of San Carlos - Talamban Campus, Cebu City, The Philippines
[6]Post-Graduate Program in Epidemiology, Federal University of Pelotas, Pelotas, RS, Brazil
[7]SAMRC/Wits Developmental Pathways for Health Research Unit, Department of Pediatrics, Faculty of Health Sciences, University of the Witwatersrand, Johannesburg, Gauteng, South Africa
[8]DSI-NRF Centre of Excellence in Human Development, University of the Witwatersrand, Johannesburg, South Africa
[9]Senior Consultant Pediatrics and Clinical Epidemiology, Sitaram Bhartia Institute of Science and Research, New Delhi, India

**Acknowledgements** The authors thank all COHORTS participants and study teams. The authors also thank Paul Melgar, the late field director of the INCAP Nutrition Trial and Longitudinal Study, in Guatemala.

**Collaborators** Additional members of the COHORTS Group include: Pelotas Birth Cohorts: Cesar G Victora; New Delhi Birth Cohort: Caroline HD Fall, Clive Osmond, Lakshmy Ramakrishnan, Bhaskar Singh and Sikha Sinha; INCAP Nutrition Supplementation Trial Longitudinal Study: Adela Sanchez; Cebu Longitudinal Health and Nutrition Survey: Isabelita Bas, Judith B Borja, Nanette R Lee, Lorna L Perez.

**Contributors** NEP and ADS conceived the idea. NEP, LSA, RM, SAP, MR-Z and ADS were involved in the conceptualisation of this analysis. LSA, MR-Z, SKB, SAB, DBC, MFK-L, BLH, NPL, MM, AMBM, SAN, LHN, LMR, HSS, FCW, ADS and the COHORTS group led and/or participated in data collection. NEP drafted the analysis plan, conducted the data management and statistical analysis, wrote the first draft of the manuscript, and incorporated edits along the review process. All coauthors reviewed and provided insightful comments to the analysis and manuscript drafts. All authors approved the final report. NEP and ADS are reponsible for the overall content as the guarantors of this work.

**Funding** The Bill and Melinda Gates Foundation (OPP1164115) funded the most recent wave of data collection in Guatemala, the Philippines, and South Africa, and data management and analysis in Brazil. Data collection in Brazil was funded by the Wellcome Trust (086974/Z/08/Z). The New Delhi Birth Cohort has also received funding from the Indian Council of Medical Research (50/1-3/TF/05-NCD-II; 3/1/2/2/15- RCH; 5/10/FR/10/2019-RCH; 5/4/8-7/2019-NCD-II), the Department of Biotechnology (BT/PR3874/MED/97/1/2011; BT/PR5317/FNS/20/552/2012), the United States National Center for Health Statistics (PL-480, RESEARCH PROJECT 0-1-658-2) and the British Heart Foundation (UKPG/05/046). South Africa (Birth to Thirty) funders are the South African Medical Research Council, Wellcome Trust (UK, 077210/Z/05/Z; 092097/Z/10/Z), University of the Witwatersrand and the DSINRF Centre of Excellence in Human Development. NEP was partially supported by a Fulbright - Colciencias Fellowship from Colombia.

**Competing interests** None declared.

**Patient and public involvement** Patients and/or the public were not involved in the design, or conduct, or reporting, or dissemination plans of this research.

**Patient consent for publication** Not applicable.

**Ethics approval** This study involves human participants and was approved by Brazil, Federal University of Pelotas, reference number 1.250.366; Guatemala, Institute of Nutrition for Central America and Panama, reference number CIE-REV-072-2017; India, Sitaram Bhartia Institute of Science and Research, reference numbers SBISR/2012/002, SBISR/RES1/3/2012, SBISR/IEC/2014/001, IEC/SBISR/2015/1 and FL/SBISR/IEC/2019–01; the Philippines, Research Ethics Committee at University of San Carlos, reference number 006/2018-01-borja; South Africa, Human Research Ethics Committee at University of Witswatersrand, reference number M180225). Parents or guardians of participants granted their written or verbal consent to participate at baseline and in childhood/adolescence follow-ups, while in adulthood participants gave their own consent. The present analysis was approved by the Institutional Review Board of Emory University (IRB number 95960). Participants gave informed consent to participate in the study before taking part.

**Provenance and peer review** Not commissioned; externally peer reviewed.

**Data availability statement** COHORTS data are not freely available due to privacy considerations. Data are available upon reasonable request to the principal investigators at each study site.

**ORCID iDs**

Natalia E Poveda http://orcid.org/0000-0001-6070-362X
Linda S Adair http://orcid.org/0000-0002-3670-8073
Reynaldo Martorell http://orcid.org/0000-0001-8862-7243
Shivani A Patel http://orcid.org/0000-0003-0082-5857
Manuel Ramirez-Zea http://orcid.org/0000-0001-5107-9175
Sonny A Bechayda http://orcid.org/0000-0003-0076-5276
Bernardo Lessa Horta http://orcid.org/0000-0001-9843-412X
Mónica Mazariegos http://orcid.org/0000-0002-2250-3683
Ana Maria Baptista Menezes http://orcid.org/0000-0002-2996-9427
Shane A Norris http://orcid.org/0000-0001-7124-3788
Linda M Richter http://orcid.org/0000-0002-3654-3192
Harshpal Sachdev http://orcid.org/0000-0002-4956-9391
Fernando C Wehrmeister http://orcid.org/0000-0001-7137-1747
Aryeh D Stein http://orcid.org/0000-0003-1138-6458

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
