## [Reviewer comments · BMJ Open]

ARTICLE DETAILS

TITLE (PROVISIONAL)	Growth patterns in childhood and adolescence and adult body composition: a pooled analysis of birth cohort studies from five low- and middle-income countries (COHORTS collaboration)
AUTHORS	Poveda Rey, Natalia Elvira; Adair, Linda; Martorell, Reynaldo; Patel, Shivani; Ramirez-Zea, Manuel; Bhargava, Santosh; Bechayda, Sonny A; Carba, Delia B; Kroker-Lobos, Maria; Horta, Bernardo; Lima, Natália; Mazariegos, Mónica; Menezes, Ana Maria; Norris, Shane; Nyati, Lukhanyo H; Richter, Linda; Sachdev, Harshpal; Wehrmeister, Fernando; Stein, Aryeh D.

VERSION 1 – REVIEW

REVIEWER	Ahuja, Kiran University of Tasmania, School of Health Sciences
REVIEW RETURNED	18-Nov-2022

GENERAL COMMENTS	The manuscript investigating the association between growth patterns in childhood and adolescence and adult body composition in LMICs presents indicates that weight gain in childhood impacts weight in adulthood. While this finding is not new, however seeing the trends from LMIC countries is a novel addition to the literature. To improve the presentation of the information, I suggest the following: Abstract: • Exposure variables need to be defined, i.e., what is meant by conditional height and conditional relative weight.• State whether the exposures were positively or negatively associated with the outcome variables in the conclusion. Introduction: • I think a clearer statement is required on the purpose of the paper, especially in comparison to the previous articles from the same study. Methods • Is there a reason why outcome variables were not used in imputation models? Results • kindly check for accuracy of data; e.g. is the mean paternal age for Brazil 1982 cohort 52.0 (7.0) years correct?• it is not clear how correlation analysis contributes to the aim of the paper Discussion • is there any explanation/speculative mechanism for the positive association between linear growth in infancy and adiposity in adulthood?• A clearer discussion of any sex differences of the associations is needed. This could include what are the possible reasons for any
--

	differences observed and how they can impact the suggested implications of the research.
REVIEWER	Poulton, Alison University of Sydney, Sydney Medical School Nepean
REVIEW RETURNED	24-Nov-2022
GENERAL COMMENTS	The researchers analysed longitudinal anthropometric data from 6 population-based cohorts from 5 low and middle income countries for predictors of adult body composition (total sample 4137). They used sophisticated statistical modelling, analysing changes in HAZ and WAZ. They used WHO standards as reference growth data. Age of measurement of adult body composition at follow-up varied from 22-46 years across cohorts, adjusting for any relevant confounders that were reported for each cohort. The proportion of men and women with adult BMI >25 varied in the different cohorts, with the lowest proportions found in South Africa and Brazil and highest in India and Guatemala for men and women respectively: note the correlation of BMI with the age of the cohort, with the oldest cohorts having the greatest proportion with BMI>25 and the highest FMI. It would be useful to comment on this. Higher birthweight predicted higher adult FMI and FFMI, but not adiposity (FM/FFM). Weight data in childhood was a greater predictor than birthweight for adult FMI, FFMI, FM/FFM, BMI and waist circumference. Associations of adult body composition were weaker for birth length than birth weight. There were no significant correlations between adult height and indices of body composition, apart from a weak correlation with abdominal circumference. The main finding appears to be the relative consistency of growth trajectories over time. It should be no surprise that more recent anthropometric measurements taken during childhood and adolescence would be more significant predictors of adult body size and composition than birth measurements. Their final sentence includes the recommendation: public health interventions should aim to promote an adequate linear growth in early life and a healthy weight gain throughout the life course to prevent the excess of adiposity in adulthood... To me this recommendation is problematic in relation to growth in height. While healthy weight gain throughout life is an appropriate recommendation, promoting adequate linear growth can only realistically be attained either by diet, which would have a greater effect on weight than height and only achieve greater height velocity at the cost of greater weight gain, or by preconceptual mate selection for greater genetic height predisposition. I could not find any statement that data in the cohorts were non-identifiable, or whether they are freely accessible in the public domain or whether an ethics application was required.

VERSION 1 – AUTHOR RESPONSE

Reviewer 1 (Dr. Kiran Ahuja, University of Tasmania)

The manuscript investigating the association between growth patterns in childhood and adolescence and adult body composition in LMICs presents indicators that weight gain in childhood impacts weight in adulthood. While this finding is not new, however seeing the trends from LMIC countries is a novel addition to the literature.

To improve the presentation of the information, I suggest the following:

1. Abstract

- Exposure variables need to be defined, i.e., what is meant by conditional height and conditional relative weight.

Response: We have added the definition of conditional growth measures in the text as follows:

“Birth weight and conditional growth (standardized residuals of height representing linear growth and of relative weight representing weight increments independent of linear size) in infancy, childhood, and adolescence”

- State whether the exposures were positively or negatively associated with the outcome variables in the conclusion.

Response: we have included some edits in the abstract conclusion as follows:

“Prenatal and postnatal relative weight gains were positive predictors of larger body size and increased adiposity in adulthood. A faster linear growth in infancy was a significant but weak predictor of higher adult adiposity.”

2. Introduction

- I think a clearer statement is required on the purpose of the paper, especially in comparison to the previous articles from the same study.

Response: We have added material to the last paragraph of the introduction to describe the innovation in this manuscript.

We have modified the text as follows:

“We expand on these previous studies by 1) analyzing for the first time the period between childhood and adolescence, 2) examining the long term-role of child growth on body composition on young and mid-adulthood (ages 22-46) given that in previous analysis some study participants were adolescents, and 3) using body composition indices (height squared adjustment) to assess body compartments (weight, fat mass, and fat-free mass) independent of linear size. Specifically, we now examine the associations of linear growth and relative weight at birth, infancy, childhood, and adolescence with measures of body size and composition (BMI, FMI, FFMI, FM/FFM ratio, and waist circumference) in young and mid-adulthood.”

3. Methods

- Is there a reason why outcome variables were not used in imputation models?

Response: We modified the imputation models to include the outcome variables and reran all the analyses. The clean version of the manuscript contains updated tables. We observed minimal changes in effect sizes and all trends remained consistent with the earlier version.

4. Results

- Kindly check for accuracy of data; e.g. is the mean paternal age for Brazil 1982 cohort 52.0 (7.0) years correct?

Response: Thank you for this careful read. This was an error, now corrected (Table 2).

- It is not clear how correlation analysis contributes to the aim of the paper

Response: Body size and body composition compartments (weight, fat mass, and fat-free mass) differ by linear size, thus, individuals who are taller tend to be heavier and might have higher fat and fat-free masses. Thus, we need to use body composition indices that control for this allometric scaling (adjustment by height squared). The use of correlation analysis is to show the statistical validity of these body composition indices, which means that the indices should be independent of their denominator (height) (1, 2).

We have added a sentence in the results section to clarify why we are presenting the correlation analysis (Lines 404-410):

“Body size and composition indices should control for allometric scaling (differences in body compartments given differences in linear size) and should be independent of their denominator (height). Thus, to assess the statistical validity of body size and composition indices, we examined the correlation between these indices with adult height. We observed that the three indices (BMI, FMI, and FFMI) were not significantly correlated with height across all study sites and sexes (correlation coefficients ranged from -0.15 to 0.10). The correlation coefficients of waist circumference with adult height were significant but weak (0.15-0.26) (Supplementary Table 14).”

References:

1. Judd SE, Ramirez-Zea M, Stein AD. Relation of ratio indices of anthropometric measures to obesity in a stunted population. *Am J Hum Biol.* 2008;20(4):446-50.
2. Wells JC, Vitora CG. Indices of whole-body and central adiposity for evaluating the metabolic load of obesity. *International journal of obesity* (2005). 2005;29(5):483-9.

5. Discussion

- Is there any explanation/speculative mechanism for the positive association between linear growth in infancy and adiposity in adulthood?

Response: We have added the following paragraphs in the discussion section:

“Linear growth is determined by the hormonal regulation of the growth hormone (GH) secreted by the pituitary and the insulin-like growth factor 1 (IGF-1), an intermediary hormone between the GH and target tissues.¹ Based on human models, growth during the first year after birth is described as an extension of fetal growth,² and linear size in infancy is influenced prominently by IGF-1 that is regulated mainly by nutritional factors than by the GH.^{3,4} Thus, IGF-1 might be the hormonal link underlying the association between faster linear growth in infancy and adult adiposity given its metabolic functions (anabolism) and role in promoting linear growth.¹ Studies of British cohorts have shown that infants (0-2 years) with a faster postnatal growth (catch-up growth) in weight or height, had higher levels of IGF-1 at age 5 years, independent of birth levels.⁵ IGF-1 levels in childhood were positively correlated with weight, height, and body composition in childhood.⁵ Besides, children who are taller tend to be heavier, and rapid weight gains and catch-up growth in postnatal years have been associated with higher adiposity and risk of obesity in childhood and adult life.⁶ In turn, animal models and clinical trials have demonstrated that excess weight causes a downregulation of the GH/IGF-1 axis leading to higher adiposity in adulthood.⁷”

Thus, increased growth in infancy, including a faster linear growth, might induce long-lasting metabolic programming of the GH/IGF-1 axis and a subsequent increased risk of adiposity. However, the evidence on the role of linear growth on obesity risk and higher adiposity later in life is limited and inconclusive.⁶ For example, a recent study of a British cohort showed that infant linear growth (ages 0-3 months) modestly improved prediction models of childhood adiposity, but

the estimates were small and with a negative trend.⁸ While, our findings showed that faster linear growth in infancy (0-24 months) has a weak but significant association with adult adiposity, showing a positive trend, in contrast to findings in children.”

References:

1. Laron Z. Insulin-like growth factor 1 (IGF-1): a growth hormone. *Mol Pathol* 2001;54(5):311-6. doi: 10.1136/mp.54.5.311
2. Karlberg J. A biologically-oriented mathematical model (ICP) for human growth. *Acta Paediatr Scand Suppl* 1989;350:70-94. doi: 10.1111/j.1651-2227.1989.tb11199.x
3. Fliesen T, Maiter D, Gerard G, et al. Reduction of serum insulin-like growth factor-I by dietary protein restriction is age dependent. *Pediatr Res* 1989;26(5):415-9. doi: 10.1203/00006450-198911000-00010
4. Ong KK, Elmlinger M, Jones R, et al. Growth hormone binding protein levels in children are associated with birth weight, postnatal weight gain, and insulin secretion. *Metabolism: clinical and experimental* 2007;56(10):1412-7. doi: 10.1016/j.metabol.2007.06.004
5. Ong K, Kratzsch J, Kiess W, et al. Circulating IGF-I levels in childhood are related to both current body composition and early postnatal growth rate. *The Journal of clinical endocrinology and metabolism* 2002;87(3):1041-4. doi: 10.1210/jcem.87.3.8342
6. Woo JG. *Infant Growth and Long-term Cardiometabolic Health: a Review of Recent Findings.* *Curr Nutr Rep* 2019;8(1):29-41. doi: 10.1007/s13668-019-0259-0 [published Online First: 2019/02/08]
7. Berryman DE, Glad CA, List EO, et al. The GH/IGF-1 axis in obesity: pathophysiology and therapeutic considerations. *Nat Rev Endocrinol* 2013;9(6):346-56. doi: 10.1038/nrendo.2013.64 [published Online First: 2013/04/09]
8. Ong KK, Cheng TS, Olga L, et al. Which infancy growth parameters are associated with later adiposity? The Cambridge Baby Growth Study. *Ann Hum Biol* 2020;47(2):142-49. doi: 10.1080/03014460.2020.1745887

- A clearer discussion of any sex differences of the associations is needed. This could include what are the possible reasons for any differences observed and how they can impact the suggested implications of the research.

Response: we have added the following paragraphs to the discussion section:

“Overall, men and women showed similar trends, larger weight gain across the life course was associated with larger body size and adiposity in adult life. However, among women, estimates for size in adolescence tended to be stronger than those for childhood, in contrast to the men, in whom childhood estimated tended to be stronger. Regarding conditional height, a faster linear growth in infancy was positively but weakly associated with higher adult adiposity among both men and women, and faster linear growth in childhood was positively associated with higher adiposity among adult women but not in men.

Sex-based dimorphism in human growth patterns, body size and composition since the prenatal period, with pronounced differences in certain age intervals is well recognized.¹⁰ In infancy, growth patterns are similar among boys and girls, a period characterized by a deceleration of linear growth and fat accretion after the first year of birth.⁹ However, in childhood, after the adiposity rebound (around age 6), there is acceleration in linear growth and weight gain, accretion that is mainly due to gains in lean mass among boys and gains in fat mass among girls.⁹ In adolescence, women reach puberty earlier and growth stops at a younger age compared to men. Thus, men have a longer prepubertal period for growth and in adolescence they experience a greater accretion in lean mass than fat mass, whereas women weight gain is characterized by a greater accretion in fat mass and their growth velocity is slower than men.^{2,9} These sex differences remain until adult life, where men and women on average have different body stature, composition, shape and fat distribution. These changes and sex-differences in body size and

composition are the result of hormonal regulations, mainly by the action of the GH/IGF-1 axis in infancy and childhood, plus the effect of steroid hormones in puberty. ^{2 9}

Thus, sex dimorphisms in growth patterns and body composition might explain the differences in the effect sizes and trends among men and women that we observed. Public health recommendations and interventions should pay special attention to these differences in growth and body composition among boys and girls mainly in childhood and adolescence, to prevent the excess of adiposity and high prevalence of obesity in adult life (higher among women), since early stages of life.”

References:

2. Karlberg J. A biologically-oriented mathematical model (ICP) for human growth. *Acta Paediatr Scand Suppl* 1989;350:70-94. doi: 10.1111/j.1651-2227.1989.tb11199.x
9. Wells JC. Sexual dimorphism of body composition. *Best Pract Res Clin Endocrinol Metab* 2007;21(3):415-30. doi: 10.1016/j.beem.2007.04.007
10. WHO Multicentre Growth Reference Study Group. WHO Child Growth Standards based on length/height, weight and age. *Acta Paediatr Suppl* 2006;450:76-85. doi: 10.1111/j.1651-2227.2006.tb02378.x

Reviewer 2 (Dr. Alison Poulton, University of Sydney)

The researchers analysed longitudinal anthropometric data from 6 population-based cohorts from 5 low- and middle-income countries for predictors of adult body composition (total sample 4137). They used sophisticated statistical modelling, analysing changes in HAZ and WAZ. They used WHO standards as reference growth data. Age of measurement of adult body composition at follow-up varied from 22-46 years across cohorts, adjusting for any relevant confounders that were reported for each cohort.

1. The proportion of men and women with adult BMI >25 varied in the different cohorts, with the lowest proportions found in South Africa and Brazil and highest in India and Guatemala for men and women respectively: note the correlation of BMI with the age of the cohort, with the oldest cohorts having the greatest proportion with BMI>25 and the highest FMI. It would be useful to comment on this.

Response: We have added a sentence in the first paragraph of the results section to incorporate the reviewer’s feedback, as follows:

“The proportion of excess weight and adiposity were correlated with the age of the cohorts, with the oldest cohorts having the greatest proportion of BMI \geq 25 kg/m² and the highest FMI. Among men, the proportion of excess weight ranged from 16.0% in South Africa to 73.2% in India, and among women ranged from 44.5% in Brazil 1993 to 85.4% in Guatemala. Participants from Guatemala, India, and Brazil 1982 had the highest FMI (women above 10 kg/m² and men above 7 kg/m²).”

2. Higher birthweight predicted higher adult FMI and FFMI, but not adiposity (FM/FFM). Weight data in childhood was a greater predictor than birthweight for adult FMI, FFMI, FM/FFM, BMI and waist circumference. Associations of adult body composition were weaker for birth length than birth weight.

Response: Thank you for this elegant synopsis of our findings. We have modified the summary of findings (first paragraph of the discussion), to highlight some points the reviewer summarizes here.

“In a pooled analysis of six prospective birth cohorts from five LMICs, we found that a higher birthweight predicted higher adult FFMI, FMI, BMI and waist circumference, but not adiposity (FM/FFM). Birth length was a positive predictor of FFMI and BMI, only among adult women. Higher relative weight gains in infancy, childhood, and adolescence were associated with higher FFMI, FMI, FM/FFM ratio, BMI and waist circumference in adult life. Associations strengthened with age, thus, effect sizes were the strongest in childhood and adolescence, almost twice the magnitude of effect sizes in infancy, and four to five times the magnitude of birth estimates. Our findings also showed that a higher conditional height in infancy was a significant but weak predictor of higher FMI and BMI among adult men and women, and higher waist circumference among women. A higher conditional height in childhood was positively but weakly associated with women’s FMI, FM/FFM ratio, and waist circumference.”

3. There were no significant correlations between adult height and indices of body composition, apart from a weak correlation with abdominal circumference.

Response: Indeed, this is what we were expecting, no correlation between body composition indices and height. The reason is that body size and body composition compartments (weight, fat mass, and fat-free mass) differ by linear size, thus, individuals who are taller tend to be heavier and might have higher fat and fat-free masses. Thus, we need to use body composition indices that control for this allometric scaling (adjustment by height squared). The use of correlation analysis is to show the statistical validity of these body composition indices, which means that the indices should be independent of their denominator (height) (1, 2).

We have added a sentence in the results section to clarify why we are presenting the correlation analysis (Lines 404-410):

“Body size and composition indices should control for allometric scaling (differences in body compartments given differences in linear size) and should be independent of their denominator (height). Thus, to assess the statistical validity of body size and composition indices, we examined the correlation between these indices with adult height. We observed that the three indices (BMI, FMI, and FFMI) were not significantly correlated with height across all study sites and sexes (correlation coefficients ranged from -0.15 to 0.10). The correlation coefficients of waist circumference with adult height were significant but weak (0.15-0.26) (Supplementary Table 14).”

References:

1. Judd SE, Ramirez-Zea M, Stein AD. Relation of ratio indices of anthropometric measures to obesity in a stunted population. *Am J Hum Biol.* 2008;20(4):446-50.
 2. Wells JC, Victora CG. Indices of whole-body and central adiposity for evaluating the metabolic load of obesity. *International journal of obesity (2005).* 2005;29(5):483-9.
4. The main finding appears to be the relative consistency of growth trajectories over time. It should be no surprise that more recent anthropometric measurements taken during childhood and adolescence would be more significant predictors of adult body size and composition than birth measurements.

Response: This is correct, our results are showing that associations strengthen with age, and this is consistent with previous studies on this topic that we reference in the discussion section.

5. Their final sentence includes the recommendation: public health interventions should aim to promote an adequate linear growth in early life and a healthy weight gain throughout the life

course to prevent the excess of adiposity in adulthood... To me this recommendation is problematic in relation to growth in height. While healthy weight gain throughout life is an appropriate recommendation, promoting adequate linear growth can only realistically be attained either by diet, which would have a greater effect on weight than height and only achieve greater height velocity at the cost of greater weight gain, or by preconceptual mate selection for greater genetic height predisposition.

Response: We agree that interventions to impact child height need to be implemented in early life, and that interventions later in childhood are likely to increase adiposity rather than stature. ^{11 12}

We have modified the text as follows:

“In summary, our findings suggest that: 1) growth in the first 1,000 days of life has a small and long-lasting association with adult body size and composition; 2) childhood and adolescence exhibit the strongest associations between relative weight gain with body size and composition in adulthood; and 3) a faster linear growth in infancy (girls and boys) and childhood (girls) predicts increased adiposity in adult life. Historical efforts to reduce growth faltering in LMICs have been successful, with little evidence to date of increases in BMI-for age.¹³⁻¹⁵ However, once nutritional and other environmental constraints on growth are removed, further nutritional inputs beyond individual requirements are likely to increase adiposity. Thus, public health interventions might focus on promoting adequate linear growth in early life and a healthy weight gain throughout the life course.”

References:

11. Adair LS, Fall CH, Osmond C, et al. Associations of Linear Growth and Relative Weight Gain During Early Life With Adult Health and Human Capital in Countries of Low and Middle Income: Findings From Five Birth Cohort Studies. *Lancet (London, England)* 2013;382(9891) doi: 10.1016/S0140-6736(13)60103-8
 12. He S, Stein AD. Early-Life Nutrition Interventions and Associated Long-Term Cardiometabolic Outcomes: A Systematic Review and Meta-Analysis of Randomized Controlled Trials. *Adv Nutr* 2021;12(2):461-89. doi: 10.1093/advances/nmaa107
 13. Martorell R. History and Design of the INCAP Longitudinal Study (1969-1977) and Its Impact in Early Childhood. *Food Nutr Bull* 2020;41(1_suppl):S8-s22. doi: 10.1177/0379572120906062 [published Online First: 2020/06/12]
 14. Kroker-Lobos MF, Ramirez-Zea M, Stein AD. Overweight and Obesity, Cardiometabolic Health, and Body Composition: Findings From the Follow-Up Studies of the INCAP Longitudinal Study. *Food and Nutrition Bulletin* 2020;41(1_suppl):S59-S68. doi: 10.1177/0379572120903222
 15. Stein AD, Wang M, Ramirez-Zea M, et al. Exposure to a nutrition supplementation intervention in early childhood and risk factors for cardiovascular disease in adulthood: evidence from Guatemala. *Am J Epidemiol* 2006;164(12):1160-70. doi: 10.1093/aje/kwj328 [published Online First: 20061003]
6. I could not find any statement that data in the cohorts were non-identifiable, or whether they are freely accessible in the public domain or whether an ethics application was required.

Response: This information is provided at the end of the manuscript under the data availability statement (Lines 571-572):

“Data availability statement COHORTS data are not freely available due to privacy considerations. Data will be available upon reasonable request to the principal investigators at each study site.”

VERSION 2 – REVIEW

REVIEWER	Ahuja, Kiran University of Tasmania, School of Health Sciences
REVIEW RETURNED	05-Feb-2023

GENERAL COMMENTS	Thank you for the updates to the paper. I have no further questions/concerns.
--

REVIEWER	Poulton, Alison University of Sydney, Sydney Medical School Nepean
REVIEW RETURNED	30-Jan-2023

GENERAL COMMENTS	The authors have made substantial improvements to the paper. I have no further comments.
---